# Informal Board Hierarchy and High-Quality Corporate Development: Evidence from China

**Haijuan Xie, Niankun Li \*, Chenglong Wang and Jiangxuan Wang**

School of Business, Guilin University of Electronic Technology, Guilin 541004, China;
17353763989@163.com (C.W.); wangjiangxuan21@163.com (J.W.)
\* Correspondence: 21052201025@mails.guet.edu.cn

**Abstract:** Promoting and realizing high-quality economic development is a major development strategy in China, but realizing high-quality macroeconomic development must be implemented at the micro-enterprise level. This paper takes Shanghai and Shenzhen A-share listed companies from 2010 to 2020 as a sample. By collecting data from CSMAR (China Stock Market & Accounting Research Database), CSMAR (China Stock Market & Accounting Research Database), and CNRDS (Chinese Research Data Services) and using a fixed-effects model, this paper investigates the impact of informal board level on the high-quality development of enterprises. The results of the study show that the informal board hierarchy contributes to the high-quality development of firms and is stronger in non-SOEs and firms with lower quality of internal control; the mechanism study shows that the informal hierarchy can contribute to the high-quality development of firms by reducing agency costs and improving business performance. In addition, higher levels of environmental uncertainty and board interaction can weaken the role of the informal hierarchy in promoting high-quality corporate development. Based on the perspective of the informal system, this paper innovatively explores how the informal level of the board of directors can overcome the shortcomings and risks of the "people" in the formal system. It is of practical significance to optimize the corporate governance structure and improve the corporate governance system and promote the high-quality development of enterprises.

**Keywords:** informal board hierarchy; agency costs; business performance; high-quality corporate development

---

## 1. Introduction

In 2017, the President pointed out in the report of the 19th Party Congress that "socialism with Chinese characteristics has entered a new era", and the central contradiction of China's society in the context of the new era has been transformed into the contradiction between people's growing need for a better life and unbalanced and insufficient development, while promoting and achieving high-quality economic development is an important development strategy to solve the problem of uneven and inadequate economic and social development. High-quality development is the development from simple pursuit of quantity and growth rate to development with quality and efficiency as the primary goal. Its basic requirements are less investment in production factors, high efficiency in resource allocation, low cost of resources and environment, and good economic and social benefits. It is an "upgraded version" of sustainable development. Since the formulation of high-quality development was put forward, the practical and academic communities have extensively researched and discussed the connotation of high-quality development and its realization mechanism. From the existing research, the research on high-quality development is mainly reflected in macroeconomic [1] industrial development [2,3] and enterprise operation [4]. There are three levels, and there is a dependency and subordination relationship among them. The improvement of enterprise operation quality promotes the development and upgrading of the industrial system, and the improvement and optimization of the industrial system promote macroeconomic restructuring and the improvement

of the total amount. Therefore, exploring how companies can achieve a high level and efficient value creation model, i.e., high-quality development at the enterprise level, is the key to achieving high-quality socio-economic development.

Ren (2020) points out that achieving high-quality economic development requires raising risk awareness and avoiding the linkage and overlap of various types of risk challenges [5]. Furthermore, enterprises also face various risk challenges in the process of continuously promoting high-quality development, and such risks often have adverse effects on the high-quality development of enterprises, and identifying, preventing, and overcoming adverse risks are effective means to effectively promote high-quality development. The existing literature on the quality development of enterprises focuses on two aspects: risks faced by enterprises and risk response. The first is the risks faced by enterprises. Both include the nature of property rights within the enterprise [6], political affiliation and financing risks [7] and operational risks [8] and external policy changes, such as tax and fee reductions [9], government subsidies [10], tax incentives [11], etc. Second is risk response. Among these are optimizing the business environment [12], promoting the marketization process [13], internal control and media attention of synergistic governance [14], and strengthening external supervision [15]. These are measures to effectively deal with the risks encountered in the high-quality development of enterprises. However, in addition to considering such macroscopic risk factors and response options, it is also important to consider the impact of the inherent uncertainty of the subjective activity of "people" on the quality development of enterprises, including whether governance is "in the same boat" or "scattered" in guiding the direction of enterprise development. There is little research in the existing literature on the mechanism of the impact of human subjectivity on the quality development of enterprises, which is the source of internal risks and the starting point for the effective prevention of internal risks in the quality development of enterprises.

As an important part of corporate governance, the board of directors has both decision-making and supervisory functions under the formal system and has a profound impact on achieving high-quality corporate development. However, the board of directors is composed of individuals, and while its effectiveness cannot be achieved without the attributes and characteristics of individuals, it also has the risks brought by people. In practice, there may be problems of insufficient allocation of energy among board members, inadequate supervision of management, conflicting decisions at meetings and difficulties in decision making, and consequently, inefficient and ineffective board decisions and inadequate supervision. It is difficult for the formal system to require or provide for the participation of individual characteristics of board members, and the disadvantages and risks of the "one-person-one-vote" system in such cases become apparent. This makes it difficult for the institutional arrangements at the governance level to be effectively brought into play in practice, thus causing artificial adverse effects on the high-quality development of enterprises. In such a context, the informal system may play a role in filling the gaps in the formal system, providing a new way of thinking and solving problems. Under the cultural background of "Seeing another better than oneself, one tries to equal him" in China, people tend to identify with people with rich knowledge backgrounds, high status levels, and strong resource control abilities and become dependent on or imitate them, forming an unequal hierarchical structure under "equality", and this phenomenon also exists in the board of directors. Based on the horizontal differences between directors' social capital, an implicit hierarchy structure is formed, which is the informal hierarchy of the board of directors. The existence of the informal board hierarchy has an impact on the subjective activities of individual directors, making the board of directors both democratic and efficient [16]. It has the dual effect of improving the efficiency of board decisions and strengthening the supervisory function, thus mitigating the risk of ineffective meetings and insufficient supervision. Therefore, the informal board hierarchy can provide a new research perspective to explain the high quality of corporate development. Although academics are aware of the governance role of the informal board hierarchy, existing

research has mainly focused on the financial performance dimension [17,18] and corporate innovation [19–22]. High-quality corporate development is closely related to efficient and scientific decision making, and the role of the informal board hierarchy in enhancing decision making is in line with this. Therefore, it is meaningful to study the impact of the informal board level on the high-quality development of enterprises from the perspective of the informal board level. Given this, this study proposes to construct a model to measure the informal board hierarchy based on the Chinese institutional context, drawing on the idea of the Gini coefficient and using the data related to the board members of A-share listed companies in Shanghai and Shenzhen from 2010 to 2020 in China to explore and reveal its mechanism and impact on the high-quality development of enterprises.

Compared with previous studies that focused more on the internal and external risks of the high-quality development of firms, this study has the following main contributions. First, it examines the impact of board risk on high-quality corporate development from the perspective of informal relationships among board members, providing a new micro perspective compared to previous studies on institutional or non-institutional arrangements at a more macro level from internal and external sources. Secondly, it explores the path mechanism of the informal level of the board of directors' role in the high-quality development of enterprises, which further improves and enriches the theoretical research related to the high-quality development of enterprises and explores more paths or possibilities for promoting the high-quality development of enterprises. Third, it examines the mechanism of the influence of internal and external environment on the role of the informal board of directors in the high-quality development of enterprises, which is of great practical significance for enterprises to correctly grasp the influence of internal and external factors in the process of promoting their high-quality development.

## 2. Theoretical Analysis and Research Hypothesis

This section summarizes the existing theories and literature on the informal board hierarchy and identifies the main ideas and hypotheses of this paper through theoretical analysis and logical derivation. What are the ways in which the informal hierarchy can influence quality development and how does this influence vary across the different internal and external contexts faced by firms? These questions will be answered in this section.

### 2.1. Analysis of the Direct Effect of the Informal Level of the Board of Directors on the High-Quality Development of the Company

According to the Company Law, director members should follow the one-person-one-vote system in handling important corporate affairs, so there is no hierarchical structure under the formal system. However, there are differences in resource control, knowledge backgrounds, ability levels, and degrees of contribution among board members, and board members are not exactly on an equal footing. Moreover, board member status creates an expectation among members, and the expected status theory suggests that this expectation usually results in a relatively high level of consensus, which creates hierarchical differences. Magee and Galinsky (2008) express a similar view, arguing that individuals make rational judgments and form consistent expectations with other board members based on the ability and level of each board member and that the more influential and capable members will achieve dominance through this expectation, which will result in a bias in psychological identification among board members, manifesting itself as a preference between different members' choices in specific matters [23]. He and Huang (2011) first proposed the definition of informal board hierarchy, which is an invisible hierarchical structure within the board of directors [18]. Ma et al. (2019) further summarize the existing studies and argue that informal hierarchy is the solidification of trust and one-way obedience order among members due to the existence of individual ability and influence differences, resulting in the status quo of high-status members in power and low-status members in cooperation [24]. Therefore, the informal hierarchy of the board of directors is because the members of the board compare the differences between each

other's capital and establish an expectation of status differences themselves, and various expectations collide in the exchange of activities and then agree in the exchange, forming a hierarchical structure under consistent expectations.

Achieving high-quality corporate development requires that companies operate according to the rules designed by the system, including a board of directors that can function effectively and improve the efficiency of decision making. However, the formal system has disadvantages that cannot be eliminated. For example, the system of independent directors can improve the corporate governance structure and strengthen the checks and balances of the board of directors if it functions effectively; however, too many independent directors taking part-time jobs will lead to energy dispersion and decision-making errors, which is not conducive to the operation and management of the enterprise [25]. In addition, the differences in the views of board members due to their own experiences, different interest representation groups, and information asymmetry among members can lead to group conflicts in board decision making. Some scholars have pointed out that differences in the division of tasks, corporate strategy, and financial views of the company are the main causes of conflicts in decision making, which may lead to a decline in the company's share price and lower investor expectations, negatively affecting the quality of corporate development. The informal hierarchy can play a unique role in addressing the above-mentioned shortcomings faced by the board of directors when making decisions under the formal system.

The mechanism of the role of the informal level of the board of directors on the quality development of the company is mainly expressed in the influence on the efficiency and scientific nature of corporate decision making, among other things. First, from the viewpoint of the relational contract, the richer the resources of the board members, the more information they can provide to the enterprise and the board, and the more they tend to occupy a higher position on the board. Xie et al. (2017) and Li et al. (2016) argue that the richness of resources and information of high-status directors brings greater value to the enterprise, and low-status boards rely on and identify with high-status directors and express their expectation and cooperation [26,27]. This abundance of resources and information also brings additional information to support the board's decision making, which in turn allows the board to make more accurate and effective decisions. With the gradual widening of the social capital gap, the hierarchical structure will also tend to be in an "orderly" state. An orderly structure is not only conducive to the exchange and transmission of information, but also enhances the willingness to cooperate of low-status members when the proposal of low-status directors is approved and supported by high-status directors. It can also promote mutual cooperation and cooperation among directors and improve the efficiency of scientific decision making. Second, from the viewpoint of power distance, as resource control, ability and influence bring a power base to high-status board members, and the respect and dependence from other members make them pay more attention to their status and influence. Shan et al. (2015) found that as the status of organizational members increases, individuals increase their motivation to exert influence on the organization and thus gain more influence [28]. Chen et al. (2020) also point out that the presence of an informal hierarchy makes high-status members reduce dissent from other members [29]. This makes it smoother for high-status directors to pursue relevant decisions, avoiding persistent ineffective arguments and quickening decision making.

Based on the above literature analysis, the informal board hierarchy can mitigate the risk of board decisions affecting the quality development of the company. In the informal hierarchy of the board of directors established based on trust and respect, board members in higher-level positions can use their authoritative positions to expand their voice, create an atmosphere of mutual respect and trust, encourage board members to communicate with each other and speak actively, which helps to alleviate the vastly different views and information asymmetry among board members, and mitigate or avoid the risk of internal conflict and contradiction among board members [18]. It further promotes the unity and stability within the board of directors and improves the communication efficiency of board

meetings [30], alleviating ineffective meetings and decision-making conflicts, overcoming potential adverse effects under human factors, and promoting high-quality corporate development. Accordingly, hypothesis H1 is proposed:

**H1.** *Informal board hierarchy is positively associated with high-quality corporate development.*

### 2.2. Analysis of the Conduction Effect of the Informal Level of the Board of Directors on the High-Quality Development of the Enterprise

The impact of the informal board hierarchy on high-quality corporate development does not happen overnight. This section considers the impact of the board's decision making and supervisory functions on corporate operations, as well as the role of the supervisory function on agency problems, and then considers how operational performance and agency costs are affected between the informal board hierarchy and high-quality corporate development.

#### 2.2.1. Transmission Effect of Business Performance

Business performance is an indicator of the results and effects achieved by an enterprise in the process of achieving its goals. In a competitive market environment, the improvement of business performance can enhance the competitiveness of an enterprise, make it better adapt to market changes and a competitive environment, and then promote the high-quality development of the enterprise, and the board of directors may act on the business performance of the enterprise to influence the high-quality development of the enterprise. First, corporate governance theory suggests that the board of directors, as the core of corporate governance, determines the company's production and operation plans and investment programs, which affects corporate performance. For example, board heterogeneity, i.e., the board size, percentage of non-executive directors, percentage of independent directors, frequency of board meetings, number of professional committees, and board incentives all contribute to the improvement of business performance by improving the quality of decision making. In particular, independent directors with technical background contribute to the innovation performance of the firm because they can provide professional advice to the firm [31]. Other scholars have also found that board composition can also affect business performance [32] and that female directors may increase tax aggressiveness [33]. Under the informal hierarchy, when a company encounters complex decisions, the reliance of low-status directors on high-status directors and the suppression of dissent by high-status directors will allow such decisions to be made by consensus within the board of directors as soon as possible, and the high-status directors can make more accurate judgments on the current situation and future development direction of the company because they have more information, so they can make decisions more quickly and scientifically and achieve the goal of promoting corporate innovation [19–22]. The effect is that of improving the financial performance of the company [18]. Secondly, resource dependence theory suggests that resource constraints are a key factor affecting the development of enterprises, and the ability of enterprises to occupy a favorable position in market competition depends on the ability to obtain and control resources. Halyna Mishchuk et al. have found that social capital has a significant impact on firm competitiveness [34]. Under the informal hierarchy, board members are ranked according to their social capital, resource possession, and ability, and high-status directors can bring more resources to the implementation of corporate decisions because they have more social resources; the abundance of resources provides more support for enterprise management and helps to improve the management level. Therefore, the informal hierarchy, as one of the manifestations of implicit characteristics within the board of directors, can act on the operational performance to improve the level of high-quality corporate development. Based on this, this paper proposes hypothesis H2:

**H2.** *The informal level of the board of directors can improve the level of quality corporate development by promoting the improvement of corporate business performance.*

2.2.2. The Transmission Effect of Agency Costs

Agency costs arise from the separation of management and ownership and refer to the need for shareholders to limit the behavior of agents through tight contractual relationships and strict monitoring of agents in order to prevent management from harming their own interests, which entails costs, i.e., through tight institutional design and hiring external audits for monitoring or imposing higher salary levels on management. According to agency theory, management will act in its interest to the detriment of the company and shareholders, generating "moral hazard" and "adverse selection" [35]. This can cause management to deviate from the business objectives of the company, which is not conducive to high-quality corporate development. As an important authority of internal corporate governance, the effectiveness of the board of directors' supervision and control over management depends on the board's ability and motivation to perform its functions. Firstly, conflict of interest theory suggests that board members have different interests among themselves, such as personal economic interests, political interests, reputation interests, etc. Differences in personal interests may influence the board's decision making and thus cause group conflicts, affect the independence and effectiveness of the board, and weaken the level of board operation. When the board of directors makes a decision that is unfavorable to the management, the management can manipulate some directors to oppose the decision, and the board of directors will form a substantial "front", which will aggravate the agency problem and the risk of management "capturing" the board of directors. This increases the risk of agency problems and management "capture" of the board. Under this risk, the informal hierarchy can harmonize the interest preferences of members, enhance collective identity and thus improve the efficiency of board operations, and reduce the occurrence of the board "in name only". Secondly, reputation mechanism theory suggests that the existence of an informal board hierarchy makes the high-status board of directors active in corporate governance activities to maintain its status and reputation and more prudent to pay attention to and actively monitor the major decisions made by the management, thus alleviating agency conflicts and reducing the risk of agency costs, ensuring that the major decisions made by the management contribute to the long-term development of the company and shareholders' interests. This ensures that management's major decisions contribute to the long-term development of the company and shareholders' interests, thus promoting the high-quality development of the company. Based on this, this paper proposes hypothesis H3:

**H3.** *The informal board hierarchy can contribute to high-quality corporate growth by curbing agency problems.*

*2.3. Analysis of the Moderating Effect of Environmental Factors on the Informal Board Hierarchy and High-Quality Corporate Development*

Companies are not in a relatively stable environment, and team decisions are influenced by a combination of external conditions and internal conditions. The influence of informal hierarchy on decision making may also vary under the influence of exogenous and endogenous factors, and this paper will explore the impact of different environmental factors on informal hierarchy and high-quality corporate development.

2.3.1. Analysis of the Impact of the External Environment

As far as the external environment is concerned, environmental uncertainty is one of the important factors that cannot be ignored. Environmental uncertainty means the unpredictability of changes in the external environment of enterprises. Enterprises cannot accurately predict the changes in the market environment faced by enterprises, which increases risks for enterprises in determining future development direction and making

strategic decisions, including the risk of making wrong judgments about the market environment and the risk of difficulty in making timely responses in the face of market changes. Under the uncertainty of the external environment, the role of the informal level of the board of directors in promoting high-quality corporate development may be enhanced or weakened, depending on whether the company relies more on the board's governance or management's decisions. Some scholars believe that the higher the uncertainty of the external environment, the more enterprises will rely on the board of directors [36], which plays a key role in the establishment of strategic decisions in order to maintain their competitiveness, and the board of directors will participate in the decision making more actively [37]. This can aid in the timely detection, adjustment, and solution of various problems in the development of enterprises, improving the innovation efficiency [38], business vitality [39], and corporate value [40]. In addition, the role of the informal hierarchy can also be effectively played to further promote high-quality corporate development. However, other parts of scholars believe that in times of environmental uncertainty, management usually has more resources and knowledge to cope with changes [41]. In addition, employees will trust management's decisions more [42]. At the same time, the uncertain environment also weakens the effectiveness of the board's supervision over management [43]. Therefore, the reduced reliance on the board of directors for high-quality corporate development and the weakening of the board's functions under environmental uncertainty will further inhibit the facilitative effect of the informal board hierarchy. Based on this, this paper proposes the following hypothesis:

**H4$_0$.** *The higher the environmental uncertainty, the stronger the contribution of the informal level of the board to the quality development of the company.*

**H4$_1$.** *The higher the environmental uncertainty, the greater the contribution of the informal level of the board of directors to the quality development of the company will not be enhanced.*

2.3.2. Analysis of the Impact of the Internal Environment

Corporate development is also dependent on the influence of the internal environment of the board of directors, and the degree of interaction within the board influences the board team's decision making [24]. However, the degree of interaction between the board of directors is uncertain as to the role of the informal level of the board of directors in promoting the high-quality development of the enterprise. On the positive side, the higher the level of board interaction, the more information exchange among board members, and the more the informal level can further strengthen the resource advantage of high-status directors and the information integration advantage of board members, improve board efficiency, accelerate scientific decision making, and promote high-quality corporate development. However, from another perspective, the number of board meetings is often an indicator of the degree of board interaction. The higher number of meetings not only indicates that the company's development faces serious problems that require frequent board meetings to discuss and solve, but also indicates that the role of high-status directors in decision making is limited and requires all members to hold frequent meetings for consultation and discussion, so the degree of board interaction reflects that the informal hierarchy is less effective in enhancing the development of the enterprise. The reduced role of decision-making efficiency weakens the positive effect on high-quality corporate development. Based on this, this paper further proposes the following hypotheses:

**H5$_0$.** *The higher the degree of board interaction, the stronger the contribution of the informal level of the board to the quality development of the company.*

**H5$_1$.** *The higher the degree of board interaction, the greater the contribution of the informal level of the board to the quality development of the company will not be enhanced.*

## 3. Study Design

This chapter contains the sources and processing of the relevant research data, as well as the model design involved in order to test the hypothesis, the definition of the relevant variables, and the rationale for their selection.

### 3.1. Sample Selection and Data Sources

This paper selects the listed companies in Shanghai and Shenzhen A-shares in China from 2010 to 2020 as the initial sample, mainly including manufacturing, finance, machinery, chemical, public utilities, medical and health, electronics, transportation, and high-tech industries. In order to make the research sample more in line with needs, the following screening was conducted. Due to the special characteristics of listed companies in the financial industry, listed companies in the financial sector were excluded from the sample; ST (special treatment: other risk warning) and *ST (special treatment: delisting risk warning) companies were excluded, and samples with missing values were excluded. Finally, 18,048 observations were obtained. Financial data, officers, directors, and company characteristics are obtained from the CSMAR database (China Stock Market & Accounting Research Database); news and press data are obtained from the CNRDS database (Chinese Research Data Services); and political background data are obtained from the CSMAR database (China Stock Market & Accounting Research Database). The data on political background are obtained from CSMAR (China Stock Market & Accounting Research Database) and annual reports and are collated manually. To eliminate outliers, all continuous variables are trimmed at 1% and 99%.

### 3.2. Variable Definition

This section contains a quantitative study of the elements of the study, i.e., the variables of interest are defined as well as the rationale for defining them in this way.

#### 3.2.1. Explained Variables

High-quality development of enterprises. There is no unanimous consensus in academic circles on how to measure the level of high-quality development of enterprises. Some scholars have used the multi-indicator method to construct an evaluation system for the high-quality development of enterprises and then calculate the development quality of each enterprise. However, the estimation results based on personal judgment may vary greatly and may not cover all the evaluation elements of the high-quality development of enterprises in a complete way. In recent years, total factor productivity (TFP) has become a popular indicator for evaluating high-quality development due to its rich information and comprehensive characteristics. Based on this, this paper selects TFP and draws on the total factor productivity (LP) measurement method of Lu and Lian (2012) as an indicator to measure the level of high-quality development of enterprises [44].

#### 3.2.2. Explanatory Variables

Informal board hierarchy. The informal hierarchy of the board is formed by the inherent horizontal differences between individual capital. It may be that the most direct and effective way to assess the informal status among the directors is to conduct direct or indirect interviews with the directors of all sample companies. However, considering the cost-effectiveness principle and the possibility of actual implementation, such an approach cannot be adopted to measure the indicators of the informal hierarchy. This paper adopts the approach of He and Huang (2011) [18] and uses the Gini coefficient to represent the clarity at the informal level of the board.

$$G = \frac{2\text{cov}\left(y, r_y\right)}{N\overline{y}} \tag{1}$$

*G* is the Gini coefficient, which measures the clarity of the informal hierarchy of the board, *y* is a measure of the social capital of the board members, and $r_y$ denotes the ranking of that status indicator in the board, while *cov* denotes the covariance between the two. N denotes the board size, and $\bar{y}$ is the mean of y. The Gini coefficient lies between 0 and 1. Near 0 means that the clarity of the informal hierarchy among board members is low, i.e., there is no clear distinction in status, and near 1 means that the differences among members are obvious in the informal hierarchy. However, established studies have different measures of y. He and Huang (2011) argue that the number of part-time positions can measure the personal capital of board members, and the higher number of part-time positions reflects their own social status [18]. Xue et al. (2021) argue that the contribution of political capital to personal and social capital is difficult to ignore in the Chinese social environment and therefore integrate the number of part-time positions and political affiliation into the measure of individual directors' social capital [20]. Chen innovatively used the number of director members' part-time positions and logarithmically calculated their effect. The media attention of board members, that is, the number of non-negative news reports of influential Chinese media such as China Securities Journal, China Business News, Securities Daily, Securities Times, Economic Observer, and 21st Century News Report were selected and logarized. The political affiliation of board members was taken as 2 for the central government, 1 for the local level, and 0 for the others, and a composite indicator was synthesized using principal component analysis to measure the status indicators of board members [29], while Liu et al. further quantified the political affiliation indicators, specifically, central = 5, provincial = 4, municipal = 3, county and district = 2, others = 1 (township and below), and 0 = no political affiliation [22]. This paper follows this approach by using principal component analysis to synthesize composite indicators and then use Gini coefficients to measure the clarity of the informal hierarchy.

### 3.2.3. Mediating Variables

Operating performance. ROA can measure the size of net profit that a company can generate per unit of assets, so this paper uses ROA to reflect the business performance status of a company.

Agency cost. In this paper, the ratio of overhead to operating income is used as a measure of agency costs for enterprises, and the higher the ratio, the greater the agency costs for enterprises.

### 3.2.4. Moderating Variables

Environmental uncertainty. Shen et al. (2012) argue that sales revenue can reflect the changes in the external environment faced by the firm [40]. Therefore, this paper follows the existing research and refers to their approach by using the sales revenue of the company in years n, n-1, n-2, n-3, and n-4 as dependent variables and 5, 4, 3, 2, and 1 as independent variables; the residual error of the model is abnormal sales revenue. Then the standard deviation of abnormal sales revenue in the past 5 years is calculated and divided by the average value of sales revenue in the past 5 years to obtain the environment uncertainty without industry adjustment. Finally, the industry-adjusted environment uncertainty is obtained by dividing the environment uncertainty without industry adjustment by the average value of environment uncertainty without industry adjustment in the same industry in the same year. The specific formula is as follows:

$$Sale = \varphi_0 + \varphi_1 Year + \varepsilon \tag{2}$$

where Sale is the sales revenue, Year is the year variable, and Year = 5 if the observation is for the current year. Year = 1 if the observation is for the past 4th year; Year = 2 if the observation is for the past 3rd year; and so on.

The degree of interaction of board members. This is measured by the number of annual board meetings.

### 3.2.5. Control Variables

Drawing on established studies, this paper selects influencing factors that may affect the high-quality development of enterprises, including company size, corporate growth, corporate value, majority shareholder ownership, nature of ownership, corporate leverage, cash flow ratio, internal control index, executive shareholding, and independent director ratio as control variables, and to mitigate the endogeneity problem, this paper uses a fixed-effects model to estimate the model while controlling for annual effects. See Table 1 for specific variable definitions and descriptions.

**Table 1.** Variable definitions and descriptions.

| Variable Type | Variable Name | Variable Symbols | Variable Definition |
|---|---|---|---|
| Explained variables | High-quality development of enterprises | TFP | Determination by LP method |
| Explanatory variables | Informal Board Hierarchy | GINI | Calculated from model (1) |
| Intermediate variables | Operating Performance<br>Agency Costs | ROA<br>AC | Net profit/total assets<br>Administrative expenses/operating income |
| Adjustment variables | Environmental Uncertainty<br>Level of Board Member Interaction | EU<br>NBM | From the model calculations (2) we get<br>Number of board meetings |
| Control variables | Company Size | SIZE | Natural logarithm of the number of employees |
| | Business Growth | GROWTH | (Operating profit for the period/operating profit for the same period last year)-1 |
| | Enterprise Value | Q | Market value/replacement cost |
| | Major shareholders' shareholding | TOP1 | Percentage of shareholding of the largest shareholder |
| | Nature of ownership | NPR | State-owned enterprises are assigned a value of 1, and non-state-owned enterprises are assigned a value of 0 |
| | Corporate leverage | LEV | Corporate gearing ratio |
| | Cash Flow Ratio | OC | Net cash flow from operating activities/total assets |
| | Executive Shareholding | MH | Several shares held by executives/total share capital. |
| | Percentage of independent directors | PD | Number of independent directors/total number |
| | Internal control | DIBO | High internal control = 1, low internal control = 0 |
| | Company | Company | Dummy Variables |
| | Annual | YEAR | Dummy Variables |

### 3.3. Research Model

In order to verify hypothesis H1, this paper constructs model (3) for benchmark regression analysis.

$$TFP_{i,t} = a_0 + a_1 GINI_{i,t} + \sum Controls + \varepsilon_{i,t} \tag{3}$$

To test hypotheses H2 and H3, this paper constructs models (4) and (5) and performs stepwise regression using models (3)–(5), where Medi is the mediating variable.

$$Medi = b_0 + b_1 GINI_{i,t} + \sum Controls + \varepsilon_{i,t} \tag{4}$$

$$TFP_{i,t} = c_0 + c_1 GINI_{i,t} + c_2 Medi + \sum Controls + \varepsilon_{i,t} \tag{5}$$

To test hypotheses H4 and H5, a moderating effect model (6) is constructed to test, where MD is the moderating variable.

$$TFP_{i,t} = d_0 + d_1 GINI_{i,t} + d_2 MD_{i.t} + d_3 GINI_{i,t} \times MD_{i,t} + \sum Controls + \varepsilon_{i,t} \quad (6)$$

In the above equation, the $a_0$–$d_0$ is a constant term, while the $a_1$–$d_1$ $c_2$–$d_2$ $d_3$ are regression coefficients of the model. Controls are a set of control variables in this paper and $\varepsilon_{i,t}$ is the random error term, where i represents an enterprise and t represents a year.

## 4. Empirical Analysis

### 4.1. Descriptive Statistics

In this paper, descriptive statistics of the main variables were conducted for the full sample, as shown in Table 2. The mean value of sample TFP is 9.26 and the standard deviation is 1.10, which indicates that there are more obvious differences in the level of high-quality development among companies, and the mean value of 9.26 still has a certain gap compared with the maximum value of 12.09. In addition, the median value of 9.16 is smaller than the mean value of 9.26, which indicates that there are more companies with a level of high-quality development below the mean value, and there is still some room for improvement in the high-quality development of listed companies. The minimum, maximum, and standard deviation of the informal board level are 0, 1, and 0.13, respectively, indicating that there is a general difference in the clarity of the informal board level among different companies, and the mean value is larger than the median, indicating that more companies have a lower level of clarity of the informal level. As for the intermediary variables, the mean and standard deviation of the business performance are 0.04 and 0.06, respectively, and the mean and maximum value of the agency cost are 0.09 and 0.05, respectively. The maximum values are 0.09 and 0.44, respectively, indicating that the sample companies have a relatively concentrated and low level of operating ability, but the overall control level of agency cost is good, and there are significant differences in the management level of different companies. The minimum value of environmental uncertainty is 0.13, the maximum value is 6.66, and the mean value is 1.30, which indicates that more companies face relatively less operating environment uncertainty and pressure. As for the number of board meetings, the minimum value of the sample is 1, and the maximum value is 58, which indicates that there is also a significant difference in the frequency of communication among board members among companies.

**Table 2.** Descriptive statistics results.

| Variables | N | Mean | Median | Sd | Min | Max |
|-----------|-----|------|--------|------|-------|--------|
| TFP | 18,048 | 9.26 | 9.16 | 1.10 | 6.80 | 12.09 |
| GINI | 18,048 | 0.33 | 0.31 | 0.13 | 0.00 | 1.00 |
| AC | 18,048 | 0.09 | 0.07 | 0.07 | 0.01 | 0.44 |
| ROA | 18,048 | 0.04 | 0.03 | 0.06 | −0.19 | 0.21 |
| NBM | 18,038 | 10.02 | 9.00 | 4.44 | 1.00 | 58.00 |
| EU | 18,048 | 1.30 | 4.44 | 1.14 | 0.13 | 6.66 |
| NPR | 18,048 | 0.43 | 0.00 | 0.50 | 0.00 | 1.00 |
| GROWTH | 18,048 | −0.07 | 0.11 | 3.49 | −20.48 | 12.66 |
| Q | 18,048 | 2.37 | 1.79 | 3.29 | 0.68 | 349.10 |
| LEV | 18,048 | 0.45 | 0.45 | 0.20 | 0.05 | 0.87 |
| TOP1 | 18,048 | 33.63 | 31.26 | 14.65 | 8.77 | 73.82 |
| MH | 18,048 | 8.75 | 0.09 | 15.45 | 0.00 | 67.43 |
| PD | 18,048 | 0.38 | 0.33 | 0.05 | 0.33 | 0.57 |
| OC | 18,048 | 0.05 | 0.05 | 0.07 | −0.16 | 0.24 |
| SIZE | 18,048 | 7.85 | 7.80 | 1.24 | 4.65 | 11.10 |
| DIBO | 18,048 | 0.56 | 1.00 | 0.50 | 0.00 | 1.00 |

### 4.2. Correlation Analysis

Table 3 shows the correlation coefficient matrix between the main variables, in which the correlation coefficient between informal board hierarchy and total factor productivity of enterprises is 0.14 and significant at a 1% significance level, which initially verified hypothesis H1. The relation coefficient between the main variables is basically below 0.5, which indicates that there is no serious problem of multicollinearity between the selected variables and can be analyzed in the next step.

**Table 3.** Correlation matrix.

| | TFP | GINI | AC | ROA | EU | NBM | LEV | SIZE | NPR | GROWTH | Q | MH | PD | OC | TOP1 | DIBO |
|---|---|---|---|---|---|---|---|---|---|---|---|---|---|---|---|---|
| TFP | 1 | | | | | | | | | | | | | | | |
| GINI | 0.140 *** | 1 | | | | | | | | | | | | | | |
| AC | −0.580 *** | −0.017 ** | 1 | | | | | | | | | | | | | |
| ROA | 0.140 *** | 0.070 *** | −0.145 *** | 1 | | | | | | | | | | | | |
| EU | −0.065 *** | −0.012 * | 0.078 *** | −0.082 *** | 1 | | | | | | | | | | | |
| NBM | 0.208 *** | 0.067 *** | −0.028 *** | −0.064 *** | 0.072 *** | 1 | | | | | | | | | | |
| LEV | 0.467 *** | 0.072 *** | −0.307 *** | −0.314 *** | 0.037 *** | 0.237 *** | 1 | | | | | | | | | |
| SIZE | 0.656 *** | 0.129 *** | −0.271 *** | 0.096 *** | −0.168 *** | 0.115 *** | 0.313 *** | 1 | | | | | | | | |
| NPR | 0.218 *** | 0.021 *** | −0.136 *** | −0.065 *** | −0.056 *** | −0.050 *** | 0.255 *** | 0.203 *** | 1 | | | | | | | |
| GROWTH | 0.070 *** | 0.007 | −0.100 *** | 0.447 *** | −0.00100 | 0.013 * | −0.067 *** | 0.019 ** | −0.022 *** | 1 | | | | | | |
| Q | −0.208 *** | 0.015 ** | 0.210 *** | 0.121 *** | 0.073 *** | −0.039 *** | −0.204 *** | −0.172 *** | −0.120 *** | 0.017 ** | 1 | | | | | |
| MH | −0.194 *** | −0.038 *** | 0.116 *** | 0.090 *** | −0.00700 | −0.00200 | −0.265 *** | −0.155 *** | −0.465 *** | 0.017 ** | 0.099 *** | 1 | | | | |
| PD | −0.002 | 0.054 *** | 0.052 *** | −0.022 *** | 0.00200 | 0.051 *** | −0.015 | −0.029 *** | −0.069 *** | −0.009 | 0.041 *** | 0.063 *** | 1 | | | |
| OC | 0.076 *** | 0.00200 | −0.094 *** | 0.396 *** | −0.092 *** | −0.132 *** | −0.182 *** | 0.157 *** | −0.032 *** | 0.105 *** | 0.044 *** | 0.024 *** | −0.012 | 1 | | |
| TOP1 | 0.225 *** | 0.058 *** | −0.174 *** | 0.108 *** | 0.0120 | −0.045 *** | 0.115 *** | 0.183 *** | 0.267 *** | 0.019 ** | −0.050 *** | −0.159 *** | 0.034 *** | 0.079 *** | 1 | |
| DIBO | 0.203 *** | 0.074 *** | −0.132 *** | 0.337 *** | −0.071 *** | −0.014 * | −0.001 | 0.145 *** | 0.070 *** | 0.198 *** | 0.001 | −0.011 | −0.011 *** | 0.120 *** | 0.096 *** | 1 |

Note: *** $p < 0.01$, ** $p < 0.05$, * $p < 0.1$.

### 4.3. Empirical Analysis of the Direct Effect of the Informal Level of the Board of Directors on the High-Quality Development of the Company

4.3.1. Baseline Regression Analysis

Column (1) of Table 4 reports the results of the regression of the model, where the GINI coefficient is 0.186 and significant at the 1% level of significance. The results of the basic regression analysis indicate that the informal board hierarchy can mitigate the risk of ineffective meetings and decision-making conflicts, improve the efficiency of board operations, and promote internal board unity and stability, and the higher the level of the informal hierarchy, the stronger the contribution to high-quality corporate development. Hypothesis 1 is verified.

**Table 4.** Baseline regression, nature of property rights, and quality of internal control subsample regression results.

| | (1) | (2) | (3) | (4) | (5) |
|---|---|---|---|---|---|
| **Variables** | **Full Sample** | **Non-State Enterprises = 0** | **State-Owned Enterprises = 1** | **Low Quality of Internal Control = 0** | **High Quality of Internal Control = 1** |
| GINI | 0.180 *** | 0.223 *** | 0.098 * | 0.231 *** | 0.123 *** |
| | (4.26) | (3.73) | (1.72) | (2.95) | (2.48) |
| Controls | YES | YES | YES | YES | YES |
| Company/Year | YES | YES | YES | YES | YES |
| Adj-R$^2$ | 0.906 | 0.907 | 0.928 | 0.914 | 0.943 |
| Differences | | 0.125 ** | | 0.108 * | |

Note: The method of testing the difference in coefficients between groups was the Fischer combination test (Permutation test), and 1000 samples were taken using the self-help sampling method (Bootstrap). *** $p < 0.01$, ** $p < 0.05$, * $p < 0.1$, t-values in parentheses adjusted for robust standard errors of clustering at the firm level, with the same below.

4.3.2. Heterogeneity Analysis

1. Heterogeneous analysis of the nature of property rights

Based on the relational contract view, the low-status board of directors is more proactive in cooperation and expects the high-status directors to play a more important role in the company's decision-making matters, and the member relationship based on trust and respect is more conducive to unity and cooperation within the organization to jointly promote the development of the company [24]. However, the informal hierarchy of the board of directors should differ between state-owned enterprises (SOEs), where some board members rely directly on direct government appointments, and non-SOEs, where all board members are elected. Board members based on political designation can, to some extent, create "ranking" behavior within the board based on power rather than status recognition, as noted by Wu et al. (2016), who argue that the desire for power and organizational rewards by lower-ranked members can lead to competition for status, which can challenge the legitimacy of those in power [27]. Thus, the role of informal hierarchy may not be conducive, and the role of informal hierarchy in SOEs should be relatively weaker than that of non-SOEs for high-quality corporate development. To verify the theoretical analysis, a group regression based on the nature of property rights and a permutation test on the coefficients are conducted. The results show that the coefficient of the informal hierarchy of SOEs is 0.125 lower than that of non-SOEs, and the difference of this coefficient is significant at a 5% level of significance, which indicates that the role of informal hierarchy is different in different natures of enterprises.

2. Internal control quality heterogeneity analysis

Although the existence of an informal board hierarchy can promote high-quality corporate development, it has also been shown that when high-status directors are self-interested, they are likely to use their superior position to suppress other dissenting voices and thus make decisions that are detrimental to the company. The level of internal control will affect the effectiveness of board decisions. Firstly, a good level of internal control will help high-status directors to be challenged by the supervisory board members present at the board meeting when they are making decisions on strategies that are detrimental to the company's growth, mitigating the possible negative effects of informal hierarchy. Secondly, major strategic decisions made by the board of directors depend on the specific implementation and cooperation of management and departments, and a good level of internal control can effectively monitor the implementation of policies and ensure that the programs are implemented and thus achieve the expected results. However, some scholars argue that a good level of internal control may inhibit the implementation of the effects of the informal level of the board of directors, and Xing et al. (2022) argue that the role of the informal level is limited by the diminishing marginal effect in the presence of high-quality internal control [45]. This may be because when the level of internal control is good, the independent directors and the supervisory board can operate effectively, and when faced with the higher board members using their prestige to dominate the opinions of internal members, they may be perceived as "ganging up" and question the normal fair and independent operation of the board. In such cases, the higher-ranking directors may limit their role to mitigate or avoid the challenge and demonstrate their independence and fairness, resulting in a limited effect of the informal hierarchy, while the higher-ranking directors may act more smoothly in implementing decisions when the quality of internal control is low. To verify what role internal control plays in it, this paper takes the Dibble internal control index as a proxy variable for the quality of internal control of the enterprise for group testing, where the mean value of the internal control index above the industry is assigned to 1; otherwise, it is 0. The group regression results are shown in Table 4 (4) and (5); it can be seen that the Gini coefficient in the group of high-quality internal control is lower than the coefficient of low-quality internal control (at 0.108) and significant at the 10% level, basically verifying the theoretical analysis that the role of informal hierarchy is limited in the case of high internal control.

### 4.3.3. Robustness Test

1.   Replace the explanatory variables

The indicator selected in this paper to measure the high-quality development of enterprises is total factor productivity calculated by the LP method, while the other two for calculating total factor productivity are the OP and OLS methods, so the OP method and total factor productivity calculated by the OLS method are selected to replace the original explanatory variables for regression analysis, and the regression results show that the GINI coefficient under the OP method is 0.182 and significant at the 1% level, and the GINI coefficient under the OLS method is 0.190 and significant at the 1% level, indicating that the findings are robust.

2.   Reduce the sample size

Due to systematic differences among the various industry samples that may affect the empirical results, Chen et al. conducted an empirical test based on a sample of manufacturing companies only when studying the effect of informal board hierarchy on director dissent [29]. For this reason, this paper draws on their sample selection method to conduct robustness tests based on a sample of manufacturing companies only and finds that the empirical results are consistent with the previous results, where the regression coefficient is 0.119 ($p = 0.000$), further verifying the robustness of the regression.

3.   Endogeneity test

(1)   Generation of lagged variables

Consider that there may be an inverse causal relationship between informal board hierarchy and high-quality corporate development, i.e., a higher level of high-quality corporate development may promote the clarity of informal hierarchy. This is because a higher level of high-quality corporate development indicates that the strategic plans and decisions made by the board of directors are correct, which may further promote the identification of low-status board members with high-status directors and further reinforce the status differences between them; therefore, this paper selects the one-period lagged GINI coefficient as the explanatory variable to regress the total factor productivity of firms under the LP method, OP method, and OLS method, respectively The regression coefficients are 0.119, 0.113, and 0.124, respectively, and all of them are still significantly positive at the 1% level, indicating that the results are robust.

(2)   Instrumental variable method

To alleviate the possible endogeneity problem, the two-stage least squares (2sls) method is chosen in this paper to address the endogeneity problem. The lagged one-period Gini and industry annual Gini means are chosen as instrumental variables, and in addition, this paper makes a relevant test of the premise of using the instrumental variables method; the Kleibergen–Paap rk Wald F statistic of 252.017 in the weak instrumental variables test is greater than the Stock–Yogo weak ID test critical values in the 10% bias critical value of 19.93, which can be judged according to the empirical criterion indicating that the instrumental variables are not weak instrumental variables; the Hansen J test *p*-value of 0.902 in the over-identification test does not reject the original hypothesis that the instrumental variables are exogenous and satisfies the prerequisites for the use of the instrumental variables method, in addition to the fact that the instrumental variables in the model are significant in the model, indicating that the results are robust.

(3)   Propensity score matching (PSM)

To mitigate the estimation bias due to the sample self-selection problem, this paper uses propensity score matching for robustness testing. The samples larger than the 50% quantile of the informal level of the board of directors are divided into the experimental group and the samples smaller than the 50% quantile are the control group, which is assigned the values of 1 and 0. In addition, the percentage of the first largest shareholder's shareholding, the number of board meetings, company leverage, company value, the percentage of independent directors, and company size are selected as covariates to calculate

the propensity score values and are matched with 1:1 nearest neighbor. Among them, the results of the balance test are shown in Table 5, and the kernel density plots before and after matching are shown in Figure 1. The results indicate that there is no significant difference in covariates between the treatment and control groups after matching, which indicates that the matching method is more reasonable. The paper further uses the matched samples for regression, and the GINI coefficient is 0.148 ($p$ = 0.01), indicating that hypothesis H1 is still supported after controlling for the sample self-selection problem, and the conclusion is more robust.

**Table 5.** Balance test results.

| Variable | Unmatched Matched | Mean Treated | Control | %Bias | %Reduct Bias | *t*-test $t$ | $p > t$ |
|---|---|---|---|---|---|---|---|
| TOP1 | U | 33.973 | 33.291 | 4.7 | - | 3.12 | 0.002 |
| | M | 33.980 | 34.338 | −2.4 | 47.4 | −1.63 | 0.103 |
| NBM | U | 10.210 | 9.827 | 8.6 | - | 5.81 | 0.000 |
| | M | 10.211 | 10.198 | 0.3 | 96.6 | 0.19 | 0.851 |
| LEV | U | 0.461 | 0.446 | 7.6 | - | 5.13 | 0.000 |
| | M | 0.461 | 0.462 | −0.3 | 95.9 | −0.21 | 0.833 |
| Q | U | 2.425 | 2.319 | 3.2 | - | 2.15 | 0.032 |
| | M | 2.366 | 2.348 | 0.6 | 82.6 | 0.63 | 0.531 |
| PD | U | 0.376 | 0.373 | 6.4 | - | 4.31 | 0.000 |
| | M | 0.376 | 0.376 | −0.0 | 99.6 | −0.02 | 0.987 |
| SIZE | U | 7.909 | 7.786 | 9.9 | - | 6.67 | 0.000 |
| | M | 7.911 | 7.910 | 0.1 | 99.2 | 0.05 | 0.958 |

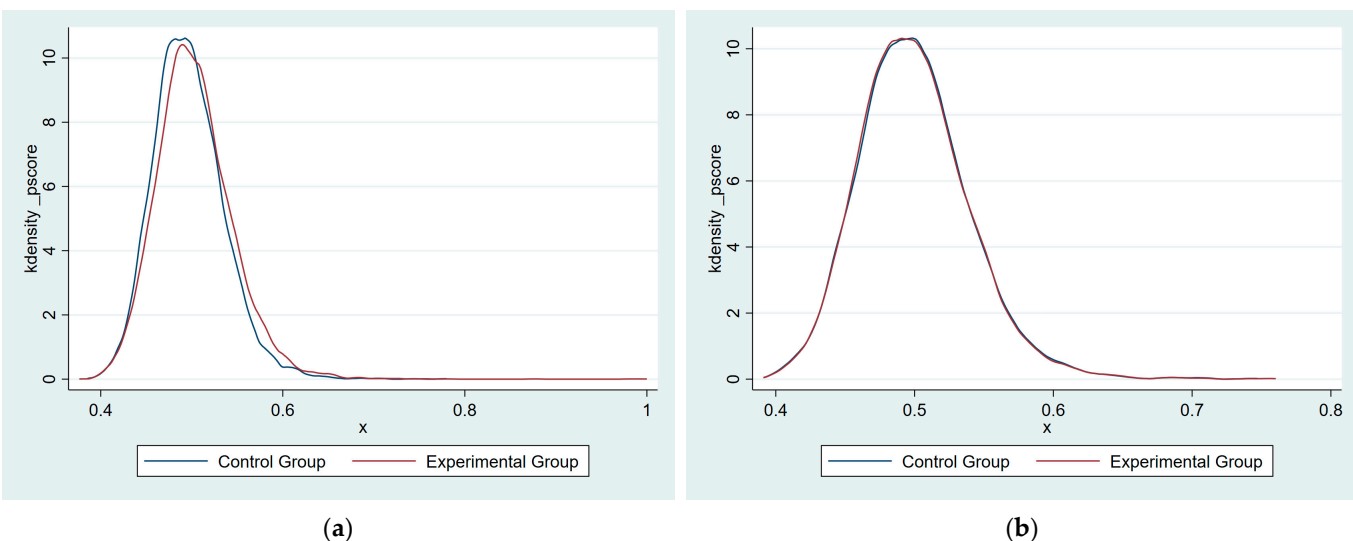

**Figure 1.** Nuclear density map. (**a**) Before matching. (**b**) After matching.

*4.4. Empirical Analysis of the Transmission Effect of the Informal Level of the Board of Directors on the High-Quality Development of the Company*

In this paper, we test the mechanism of the role of the informal level of the board of directors in influencing the high-quality development of the firm through a mediating effect model, here drawing on the study by Wen et al. (2014) [46], using the stepwise regression method, where (1) in Table 6 is a test of the main effect, (2) and (5) are tests of the explanatory variables on the mediating variables, respectively, and (4) and (7) are the explanatory variables included in the regression model on the explanatory variables together with the mediating variables. From (5), it can be seen that the GINI coefficient is significant and positive at the 0.01 level of significance, indicating that the informal level of the board of directors can significantly contribute to the improvement of business

performance, and the coefficient of ROA in (7) is significantly positive, proving that the informal level of the board of directors can contribute to the high quality of business development by improving business performance. In addition, the GINI coefficient of (2) is significantly negative at the 5% significance level, indicating that the informal board level effectively suppresses the agency problem, and the agency cost coefficient of (4) is also significantly negative at the 1% level, confirming that the informal board level can contribute to the high-quality development of the firm by suppressing the agency problem. The GINI coefficients in (4) and (7) are significantly positive and decrease to different degrees compared to the main effect, indicating that business performance and agency costs play a partially mediating role. In summary, hypotheses H2 and H3 are confirmed. To further verify the robustness of the mediation effect, this paper adopts the Sobel method of testing the mediation effect, and the results are shown in Table 7, which further proves the robustness of the conclusion of the mediation mechanism analysis.

**Table 6.** Informal board levels, business performance, agency costs, and high-quality corporate development.

| | (1) | (2) | (3) | (4) | (5) | (6) |
|---|---|---|---|---|---|---|
| **Variables** | **TFP** | **AC** | **TFP** | **ROA** | **TFP** | **TFP** |
| GINI | 0.180 *** | −0.010 ** | 0.134 *** | 0.014 *** | | 0.146 *** |
| | (4.26) | (−2.26) | (3.82) | (4.14) | | (3.57) |
| AC | | | −4.657 *** | | | |
| | | | (−26.06) | | | |
| ROA | | | | | 2.392 *** | 2.374 *** |
| | | | | | (17.26) | (17.28) |
| LEV | 0.618 *** | −0.026 *** | 0.496 *** | −0.109 *** | 0.880 *** | 0.876 *** |
| | (7.41) | (−3.47) | (7.04) | (−19.20) | (10.62) | (10.60) |
| SIZE | 0.345 *** | −0.009 *** | 0.302 *** | 0.005 *** | 0.335 *** | 0.334 *** |
| | (14.07) | (−5.64) | (13.93) | (3.44) | (14.07) | (14.07) |
| NPR | −0.063 | 0.000 | −0.062 | −0.005 | −0.051 | −0.050 |
| | (−1.16) | (0.02) | (−1.45) | (−1.52) | (−0.99) | (−0.97) |
| GROWTH | 0.009 *** | −0.002 *** | 0.002 ** | 0.005 *** | −0.003 *** | −0.003 *** |
| | (8.91) | (−11.05) | (2.34) | (33.41) | (−2.98) | (−2.86) |
| Q | −0.007 * | 0.001 * | −0.002 | 0.001 | −0.009 * | −0.009 * |
| | (−1.92) | (1.83) | (−1.50) | (1.29) | (−1.72) | (−1.72) |
| MH | 0.000 | 0.000 | 0.001 | 0.000 *** | −0.001 | −0.001 |
| | (0.04) | (0.98) | (0.43) | (3.45) | (−0.47) | (−0.67) |
| PD | 0.142 | −0.003 | 0.128 | −0.005 | 0.167 | 0.155 |
| | (0.88) | (−0.22) | (0.90) | (−0.38) | (1.08) | (0.99) |
| OC | 0.644 *** | −0.054 *** | 0.393 *** | 0.120 *** | 0.356 *** | 0.358 *** |
| | (7.44) | (−5.91) | (5.53) | (13.25) | (4.24) | (4.26) |
| TOP1 | 0.000 | −0.000 | −0.000 | 0.000 *** | −0.001 | −0.001 |
| | (0.13) | (−0.61) | (−0.08) | (4.72) | (−0.71) | (−0.69) |
| DIBO | 0.116 *** | −0.008 *** | 0.078 *** | 0.017 *** | 0.077 *** | 0.077 *** |
| | (14.48) | (−10.11) | (11.60) | (21.80) | (9.87) | (9.89) |
| Constant | 5.792 *** | 0.178 *** | 6.621 *** | 0.026 * | 5.794 *** | 5.730 *** |
| | (28.87) | (12.82) | (36.79) | (1.91) | (29.75) | (29.44) |
| Observations | 18,048 | 18,048 | 18,048 | 18,048 | 18,048 | 18,048 |
| Adj-R$^2$ | 0.906 | 0.711 | 0.930 | 0.628 | 0.911 | 0.911 |
| FIRM FE | YES | YES | YES | YES | YES | YES |
| Year FE | YES | YES | YES | YES | YES | YES |

*** $p < 0.01$, ** $p < 0.05$, * $p < 0.1$.

**Table 7.** Robustness test results of the Sobel method for mediating effects.

| Intermediate Variables | Statistic | Intermediary Effect as a Percentage |
|---|---|---|
| Agency Costs | 2.331 | 0.355 |
| Operating Performance | 4.154 | 0.261 |

### 4.5. Empirical Analysis of the Moderating Effect of Environmental Factors on Informal Board Levels and High-Quality Corporate Development

As mentioned earlier, enterprises are affected by both the external environment and their conditions in the process of development, and it is necessary to comprehensively examine the impact of the informal hierarchy on the high-quality development of enterprises under internal and external conditions, so this paper introduces the environmental uncertainty index as the measurement variable of external conditions index and the number of board meetings as the measurement variable of internal conditions, respectively, including both of them into the model as moderating variables for analysis. To avoid the collinearity problem that may be caused by the cross-term being included in the regression model, the cross-term is centralized in this paper. The specific regression results are shown in Table 8.

**Table 8.** Informal board levels, environmental uncertainty, number of board meetings, and quality corporate development.

| | (1) | (2) | (3) |
|---|---|---|---|
| **Variables** | **TFP** | **TFP** | **TFP** |
| GINI | 0.180 *** | 0.175 *** | 0.184 *** |
| | (4.26) | (4.18) | (4.36) |
| EU | | 0.044 *** | |
| | | (6.42) | |
| NBM | | | 0.007 *** |
| | | | (3.10) |
| GINI*NBM | | | −0.014 ** |
| | | | (−2.28) |
| GINI*EU | | −0.062 * | |
| | | (−1.70) | |
| LEV | 0.618 *** | 0.599 *** | 0.604 *** |
| | (7.41) | (7.29) | (7.32) |
| SIZE | 0.345 *** | 0.340 *** | 0.342 *** |
| | (14.07) | (14.16) | (13.98) |
| NPR | −0.063 | −0.051 | −0.060 |
| | (−1.16) | (−0.95) | (−1.11) |
| GROWTH | 0.009 *** | 0.009 *** | 0.009 *** |
| | (8.91) | (8.66) | (8.80) |
| Q | −0.007 * | −0.007 * | −0.007 * |
| | (−1.92) | (−1.92) | (−1.90) |
| MH | 0.000 | 0.000 | 0.000 |
| | (0.04) | (0.21) | (0.08) |
| PD | 0.142 | 0.187 | 0.139 |
| | (0.88) | (1.16) | (0.88) |
| OC | 0.644 *** | 0.656 *** | 0.666 *** |
| | (7.44) | (7.67) | (7.74) |
| TOP1 | 0.000 | −0.000 | 0.000 |
| | (0.13) | (−0.36) | (0.12) |
| DIBO | 0.116 *** | 0.115 *** | 0.115 *** |
| | (14.48) | (14.55) | (14.48) |
| Constant | 5.792 *** | 5.780 *** | 5.757 *** |
| | (28.87) | (29.49) | (28.86) |
| Observations | 18,048 | 18,048 | 18,048 |
| Adj-$R^2$ | 0.906 | 0.906 | 0.907 |
| FIRM FE | YES | YES | YES |
| Year FE | YES | YES | YES |

*** $p < 0.01$, ** $p < 0.05$, * $p < 0.1$.

Firstly, from the external environment analysis, the cross-product term of environmental uncertainty index with environmental uncertainty index and informal hierarchy is included in the benchmark regression model, and the main explanatory variables are still significantly positive at a 1% significance level, and the coefficient of environmental

uncertainty index is also significantly positive at 1% significance level, indicating that the higher the external environmental uncertainty faced by enterprises, the better the quality of enterprise development. This is because enterprises are more motivated to improve their development strategies and business conditions to better adapt to the changes in the external environment when they face a certain degree of complex external conditions, thus promoting the transformation and upgrading of enterprises. However, we can see from the regression results that the coefficient of the cross-product term is negative, i.e., environmental uncertainty weakens the role of the informal level of the board of directors in promoting high-quality corporate development, which verifies hypothesis $H4_1$ that when companies face a complex and uncertain external environment, management can play a greater role than the board of directors, and corporate development relies more on the decisions of management than governance, resulting in a weaker role of the informal level in promoting high-quality corporate development. The role of the informal level in promoting high-quality development is weakened.

In addition, the interaction term between informal hierarchy and the number of board meetings in Table 8 is significantly negatively correlated at the 5% level, but the coefficients of the number of board meetings and informal hierarchy are positively correlated with total factor productivity at the 1% level, indicating that the higher the degree of board member interaction, the weaker the contribution of informal hierarchy to high-quality corporate development, which verifies hypothesis $H5_1$. Although a higher number of meetings can pool ideas and promote information sharing among board members, the informal hierarchy arises precisely because of the differences in status and competence among board members, while the informal hierarchy can take advantage of the hierarchical position of high-status directors to accelerate the efficiency of internal decision making. However, more frequent meetings may reflect the weak cohesiveness of the high-status directors or the "disagreement" of other board members with the high-status directors, resulting in the inability to use their advantages to gather basic consensus and the need for frequent meetings to negotiate and make decisions, which makes it difficult to achieve the original effect and level of the informal hierarchy and weakens the role of promoting high-quality corporate development.

## 5. Summary

This chapter contains a brief summary and conclusion of all the above, mainly including conclusion, insight, and discussion.

### 5.1. Conclusions

Using the data of listed companies from 2010 to 2020, this paper empirically tests the relationship between the informal level of the board of directors and high-quality corporate development and explores the mechanism of action and the path of influence between the informal level of the board of directors and high-quality corporate development. The results show that the informal board hierarchy can positively promote the level of high-quality corporate development by mitigating decision risk and supervision risk, and the promotion effect of informal hierarchy varies among companies with different property rights and internal control quality levels. The informal layer of the board of directors can promote high-quality corporate development through the path of improving business performance and suppressing agency problems. When the uncertainty of the external environment is higher and the interaction level of board members is stronger, the role of the informal board hierarchy in promoting high-quality corporate development will gradually diminish.

### 5.2. Insights

1.  Properly grasp and utilize the role of informal hierarchy in promoting high-quality corporate development

As an informal system, the informal level of the board of directors plays a positive role in mitigating or overcoming the risks that cannot be avoided by the formal system,

such as the inefficient board of directors and insufficient supervision, and has a positive significance in maintaining a healthy and stable functioning of the governance level and promoting high-quality corporate development, so the favorable factors of the informal level should be actively guided and utilized to improve corporate governance. However, the level of informal layers is generally low in the sample companies, so companies need to pay attention to and guide the role of informal layers in the board of directors to improve their governance effectiveness.

2.  The role of the informal hierarchy in the quality development of enterprises needs to consider the impact of the different environments in which enterprises are located.

The role of the informal hierarchy varies from company to company, and companies with a lower-level role can adjust their corporate policies and change the environment they face to further promote their beneficial effects. In addition, existing companies do not pay attention to the coordination of the informal level with the formal system such as internal control and board meetings, and the informal system and the formal system appear to be growing in opposite directions, so companies need to pay attention to the combination of various factors that are beneficial to the quality development of the company to play a synergistic role and to avoid the occurrence of mutual substitution and the situation of the two.

*5.3. Discussion*

This study further verifies the positive role of the informal board hierarchy in corporate governance compared to the existing one, and its results are similar to those of Xue and Wang et al. [19,20], i.e., they affirm the positive role of the informal board hierarchy in corporate development, but they are also different from those of Chen and Xie et al. [26,29], who found that the informal hierarchy can suppress internal dissenting voices and may result in the negative consequences of power concentration. This is where this paper falls short, i.e., it is unable to detect and measure whether there are potential adverse effects of self-interested behavior of high-status directors on corporate development. In addition, the social capital preferred by different directors is different, and it may not be possible to fully measure the hierarchy within the board based on part-time positions, political affiliations, and news reports alone, which is an area that needs further refinement and improvement in the future.

**Author Contributions:** Writing—original draft, H.X. and N.L.; Writing—review & editing, C.W.; Supervision, J.W. All authors have read and agreed to the published version of the manuscript.

**Funding:** This research was funded by National Social Science Foundation of China (22XGL008) and Guangxi Philosophy and Social Science Planning Research Project (20FGL027).

**Institutional Review Board Statement:** Not applicable.

**Informed Consent Statement:** Not applicable.

**Data Availability Statement:** As the data of this paper involves the dissertation of the ungraduated, it will not be disclosed for the time being. If you have any questions, please contact the corresponding author to obtain the relevant data.

**Conflicts of Interest:** The authors declare no conflict of interest.

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
