# Peer review of "Informal Board Hierarchy and High-Quality Corporate Development: Evidence from China"

_sustainability, doi:10.3390/su15118914_

Round 1

Reviewer 1 Report

Comments and Suggestions for Authors in Attachment. 

I have no serious comments on quality of English language. 

Author Response

Dear Reviewer,

       Thank you for reviewing my paper in your free time. I admire your academic rigor, especially your high academic attainment, as I can see from the papers you have reviewed. I have made corrections through your review comments, and I would like to explain the corrections here, and express my gratitude to you again.

Comments1:

Chapters: 2., 2.2., 3., 3.2., 4. and 5. are divided into further subchapters, but the text is missing. Each chapter and subchapter should contain text.

Answer1:

I have added textual content under this section, thank you for your criticism and correction.

Comments2:

Line 251 - authors state: “Under the informal hierarchy, board members are ranked according to their social capital, ....“. However, it is not clear what specifically the authors consider important within the framework of social capital and with regard to the addressed area. What should social capital include? What should a mechanism for creating an informal hierarchy look like? It would also be interesting to present in a clear form the basic differences between formal and informal hierarchy from the authors' point of view.

Answer2:

     Social capital refers to the resources possessed by a certain person, such resources include the ability to obtain information, the level of knowledge possessed and including human connections, etc., but the importance is different for each person. In a certain group, each director has different evaluation criteria, for example, A has admiration for high intellectual B, but this knowledge resource may be insignificant compared to other possible resources possessed by B. However, A only uses this as its evaluation criterion for B. This is a preference specific to A. By analogy, everyone also has different preferences, so it may not specifically refer to a certain capital particularly important. The informal hierarchy of the board of directors is also formed because of this preference, which comes from one person's trust and reliance on something specific to another person, although everyone has different criteria for judging, but the members of the group are communicating and comparing with each other, and the hierarchy will be formed under the agreement of various comparisons and exchanges. Especially in the context of Chinese "Confucian culture", in the board of directors, the younger board members have a special respect for the older directors, which is the result of cultural influence. stable standard, and a hierarchical structure is formed.

Comments3:

In subchapter 2.2.2. authors focused on "agency costs". It would be appropriate to characterize in more detail what the mentioned costs mean and what they

include.

Answer3:

I have added textual content under this section, thank you for your criticism and correction.

Comments4:

There is a similar comment regarding line 323 - "environmental uncertainty". It is a very broad term, and therefore it would be appropriate to specify it.

Answer4:

I have added textual content under this section, thank you for your criticism and correction.

Comments5:

 H4a and H4b; H5a and H5b – from the point of view of the methodological procedure of creating hypotheses, the authors should determine which hypothesis is basic (null) and which hypothesis is alternative.

Answer5:

I have corrected this part of the content, thank you for your criticism and correction

Comments6:

Abbreviations are used on lines 362, 365 and 366. I recommend authors to include the full meaning in parentheses

Answer6:

I have added textual content under this section, thank you for your criticism and correction.

Comments7:

Within the research sample, it is appropriate to indicate specifically which companies were included in the research (e.g. sector / segment) - a more detailed description of the research sample of companies from various points of view.

Answer7:

I have added textual content under this section, thank you for your criticism and correction.

Comments8:

Chapter "5. Conclusion and Insights" contains a summary and repetition of the conclusions from the research and chapter 4. Based on the IMRAD structure, I recommend supplementing the "Discussion" chapter for the purpose of comparison with the results of other authors. It is also necessary to supplement the research limits. This part is completely missing.

Answer8:

I have added this section, thank you again.

I hope my reply will not cause you any trouble, and I wish you good work and good health.

Yours sincerely,

Li Niankun

Address: Guilin University of Electronic Technology

Reviewer 2 Report

The paper is interesting and devoted to important issues of corporate development. The dataset is convincing. The methods are aligned with the dataset and research hypotheses. 

Some comments should be considered by authors before publication as in the given form the paper contains some inconsistencies:

(1) the literature review is performed mainly using outdated sources which is surprising for this topic. The recent sources are rare in the reference list and they are just briefly used in hypotheses justification. So, it will be appropriate if authors pay attention to related issues to the board governance, particularly, investigated in papers:

Farhan, N. H. S., Tabash, M. I., AlMaqtari, F. A., & Yahya, A. T. (2020). Board composition and firms’ profitability: Empirical evidence from pharmaceutical industry in India. Journal of International Studies, 13(3), 180-194. doi:10.14254/2071- 8330.2020/13-3/12

Mishchuk, H., Štofková, J., Krol, V., Joshi, O., & Vasa, L. (2022). Social Capital Factors Fostering the Sustainable Competitiveness of Enterprises. Sustainability, 14(19), 11905.  https://doi.org/10.3390/su141911905

Cortellese, F. (2022). Does the gender composition of the board of directors have any effect on tax aggressiveness in western countries?. Economics and Sociology. Economics and Sociology, 15(1), 11-22. doi:10.14254/2071-789X.2022/15- 1/1

(2) section 3 should be improved. The selection of data requires a more detailed explanation. Particularly, the sources to find the mentioned listed companies can be clear for readers from China, but the sources of this information are unavailable for international readers. The authors should clarify the source of financial information used for calculations clearly. Stock exchange archives? public list of companies? other source - it should be mentioned in the text.

Besides, it seems that this paper is part of the bigger report. Some parts are comprehensive. Particularly, authors should explain abbreviations like "ST and *ST companies", "CSMAR", "CNRDS" when mentioning them at the first time.

(3) I find the title "Correlation coefficient matrix" incorrect. Just the Correlation matrix is enough.

(4) The text should be carefully revised. There are some technical inconsistencies like capital letters in the middle of the sentences (see, for example, lines 360, 745). The sentence in lines 358 - 362 is incomprehensive, it should be divided. Formulas should be placed at the center. Please, revise the text.

(5) "Insights" is rather a discussion. This section is omitted and according to the authors' findings, it will be useful to add some discussions. The comparisons with results obtained by other authors are required in this section, not only the generalization of the own results. 

In general, the English is fine, however, some corrections are needed - see comment 4.

Author Response

Dear Reviewer,

       Thank you for reviewing my paper in your free time. I admire your academic rigor, especially your high academic attainment, as I can see from the papers you have reviewed. I have made corrections through your review comments, and I would like to explain the corrections here, and express my gratitude to you again.

Comments1:

       The literature review is performed mainly using outdated sources which is surprising for this topic. The recent sources are rare in the reference list and they are just briefly used in hypotheses justification. So, it will be appropriate if authors pay attention to related issues to the board governance, particularly, investigated in papers:

Answer1:

       Thank you for your criticism. The reason why my references are a bit old is that there are not enough Chinese studies on board governance, I started my research based on the current situation of Chinese studies, so I did not study enough on other aspects.

Comments2:

       Section 3 should be improved. The selection of data requires a more detailed explanation. Particularly, the sources to find the mentioned listed companies can be clear for readers from China, but the sources of this information are unavailable for international readers. The authors should clarify the source of financial information used for calculations clearly. Stock exchange archives? public list of companies? other source - it should be mentioned in the text. Besides, it seems that this paper is part of the bigger report. Some parts are comprehensive. Particularly, authors should explain abbreviations like "ST and *ST companies", "CSMAR", "CNRDS" when mentioning them at the first time.

Answer2:

I have added textual content under this section, thank you for your criticism and correction.

Comments3:

I find the title "Correlation coefficient matrix" incorrect. Just the Correlation matrix is enough.

Answer3:

I have corrected this part of the content, thank you for your criticism and correction.

Comments4:

The text should be carefully revised. There are some technical inconsistencies like capital letters in the middle of the sentences (see, for example, lines 360, 745). The sentence in lines 358 - 362 is incomprehensive, it should be divided. Formulas should be placed at the center. Please, revise the text.

Answer4:

I am sorry for my lack of care, and I have corrected this error.

Comments5:

"Insights" is rather a discussion. This section is omitted and according to the authors' findings, it will be useful to add some discussions. The comparisons with results obtained by other authors are required in this section, not only the generalization of the own results.

Answer5:

I have added textual content under this section, thank you for your criticism and correction.

I hope my reply will not cause you any trouble, and I wish you good work and good health.

Yours sincerely,

Li Niankun

Address: Guilin University of Electronic Technology

Reviewer 3 Report

Dear authors

Although the paper provides important results, the structure of the content needs several adjustments are:

·         The abstract does not explain the research methodology and data collection method, so the abstract should be rearranged to present the study methodology and originality of the research, as well as the potential applied benefits of the results.

·         The structure of the research needs to be rearranged to clarify the logical sequence of ideas, with the need to develop the hypothesis-building section in a manner appropriate to the subject of the study.

Author Response

Dear Reviewer,

       Thank you for reviewing my paper in your free time. I admire your academic rigor, especially your high academic attainment, as I can see from the papers you have reviewed. I have made corrections through your review comments, and I would like to explain the corrections here, and express my gratitude to you again.

Comments1:

       The abstract does not explain the research methodology and data collection method, so the abstract should be rearranged to present the study methodology and originality of the research, as well as the potential applied benefits of the results.

Answer1:

       Thank you for pointing out the problem. I have adjusted in this part. Thank you very much.

Comments2:

      The structure of the research needs to be rearranged to clarify the logical sequence of ideas, with the need to develop the hypothesis-building section in a manner appropriate to the subject of the study.

Answer2:

I'm sorry, but I'm not sure I fully understand what you mean. I'd like to start with some of the ideas for hypothesis construction in this article: Firstly, the direct influence of informal hierarchy on the high-quality development of enterprises is proposed, and then the mediating effect of business performance and agency cost is considered respectively from the decision-making function and supervision function of the board of directors. In addition, considering the comprehensive influence of internal and external conditions on the decision-making of the board of directors, the different influence of internal and external factors on the decision-making effect of informal hierarchy is further proposed. If you have any questions, please provide further detailed introduction, so that I can further improve the article. Thank you for very much.

I hope my reply will not cause you any trouble, and I wish you good work and good health.

Yours sincerely,

Li Niankun

Address: Guilin University of Electronic Technology

Reviewer 4 Report

Please see attached Referee Report.

Author Response

Dear Reviewer,

       Thank you for reviewing my paper in your free time. I admire your academic rigor, especially your high academic attainment, as I can see from the papers you have reviewed. I have made corrections through your review comments, and I would like to explain the corrections here, and express my gratitude to you again.

Comments1:

The first major concern relates to the main variable of interest, GINI, and what the authors are trying to measure. The regression models all include Firm fixed effects as commonly done in corporate finance to control for time-invariant effects for firms. However, there has yet to be any indication that the informal hierarchy of the firms tends to change or does change. Thus, if the informal hierarchy of the firm does not change, it is being accounted for in the firm fixed effects, and the GINI variable cannot capture the informal hierarchy. There would need to be rationale and documentation that the informal hierarchy changes occur in the sample. Additionally, GINI does not measure the informal hierarchy of the board but attempts to measure the clarity of the informal hierarchy. These are two different items. The authors need to decide whether they are testing whether the informality of the board hierarchy impacts the corporation or if the clarity of (or how clear) the informal board hierarchy impacts the corporation.

Answer1:

Thank you very much for asking this question. The informal hierarchy does change, and the composition of the board of directors varies from company to company, as does the social capital owned, so the informal hierarchy calculated according to the formula will change accordingly. Also, it is subject to change as members move within the board, i.e., new members join and old comrades exit. In addition, GINI is indeed a measure of the clarity of the informal hierarchy, meaning that the larger the indicator, the clearer the ranking of each within the board group will be, but it also has shortcomings, such as it may not be a true representation of the actual level of the informal hierarchy. However, the research on informal hierarchy in China, including the top Chinese journal "Management World", also relies on this measure, and this method is also based on the research results of the journal "Management World", but there is no better measure of this aspect yet. I am sorry to say that I have no choice but to expand on this research. If you have a better idea, please let me know, and I appreciate your help.

Comments2:

The second major concern relates to the main findings in Table 4 (identified by the authors).

These models should be adjusted to include the entire sample and incorporate dummy variables to capture the effects of the ownership structure and the internal quality control. These can also interact to conclude the impact observed for state-owned enterprises with high internal quality control. This would be a better specification and provide more insight.

Answer2:

Your question is very relevant and demonstrates your excellent statistical foundation, which I admire very much and hope to have the opportunity to communicate with you in depth in the future. For example, there are a large number of companies in different industries in the sample, such as manufacturing, real estate, utilities, healthcare, etc. There is significant heterogeneity in different industries or companies, and group regression can be used to divide the data set into multiple subsets (i.e., "groups") and fit a separate linear regression model to each subset. A separate linear regression model is fitted for each subset. This approach may be more applicable to the presence of multiple heterogeneous subgroups in this sample, each with different characteristics and influencing factors. By using grouped regression, we can more accurately capture the differences between subgroups and avoid overfitting the entire data set. Therefore we might consider still keeping the original subgroup regression unchanged. I have benefited from the very useful suggestions you made and am considering using the methods you mentioned in future studies.

Comments3:

I appreciate the authors acknowledging the potential endogeneity issue with the research.

However, the regression results for the instrumental variable approach would provide more

compelling evidence than the base regression analysis in Table 4. Table 5 provides evidence that the propensity score matching does a good of matching. Also, I would like to draw the author’s attention to the fact that Tables 6 and 8 would be exposed to this same endogeneity issue and thus should be incorporated into the analysis. Furthermore, Table 4 shows differential impacts of GINI for the TFP variable and is absent in Tables 7 and 8. It would be easy to argue that firms with high-quality internal controls would also impact TFP, meaning the model is subject to omitted variable bias. Finally, the authors repeatedly state that variables are significant at certain p-values, but there is no discussion on the economic interpretation or significance.

Answer3:

Thank you very much for pointing out the possible endogeneity problem in Table 6 and Table 8, but probably because of the direction of research in my country, almost all papers, including those published in top Chinese journals, hardly see endogeneity tests for moderating and mediating effects, and almost all of them only test for main effects to examine the correctness of the underlying findings. Therefore, I do not yet have a better method or empirical reference to verify the possible endogeneity issues in Tables 6 and 8, and can only follow the established research practice and combine it with a fixed effects model to mitigate the possible estimation bias and endogeneity issues as much as possible. Your comments are very important and I will ask more relevant scholars in my later research and do my best to make more people focus on empirical rigor as you do. Also in view of the possible impact of the identified internal controls on the high quality development of the firm, I have included them as control variables and re-estimated all regression models and thank you again for your critical corrections.

Comments4:

Table 4 needs to replace “Control” with “Yes” to be similar to other tables. It also needs

to use the different fixed effects notation as done in other tables.

Answer4:

I am very sorry, it was an oversight on my part and I have now corrected it. Thank you for your criticism and correction.

Comments5:

Figure 2 needs to be changed to English.

Answer5:

Thanks for pointing out that the new version has been uploaded.

Comments6:

Do you find it surprising that NPR does not load on any models in Tables 6 and 8?

Answer6:

Since the variables are kept to three decimal places, NPR does not show the latter digits due to the very small value, so it shows 0.

Comments7:

Is NBM an annual value? I ask as the maximum is 58 meetings, which is a lot for a year.

Answer7:

This is an annual figure, and I was a bit surprised by this, so I went to check the relevant information and found that this is an A-share listed company with the company code 600708 and the company name Bright Real Estate Holdings Limited, whose company report disclosed to the public in 2018 stated that it held 58 meetings, and the following is a screenshot of the relevant report:

I hope my reply will not cause you any trouble, and I wish you good work and good health.

Yours sincerely,

Li Niankun

Address: Guilin University of Electronic Technology

Round 2

Reviewer 2 Report

Changes are sufficient. Recommended to be published.